# CONTINUAL CONTRASTIVE SPOKEN LANGUAGE UNDERSTANDING

## ABSTRACT

Recently, neural networks have shown impressive progress across diverse fields, with speech processing being no exception. However, recent breakthroughs in this area require extensive offline training using large datasets and tremendous computing resources. Unfortunately, these models struggle to retain their previously acquired knowledge when learning new tasks continually, and retraining from scratch is almost always impractical. In this paper, we investigate the problem of learning sequence-to-sequence models for spoken language understanding in a class-incremental learning (CIL) setting and we propose COCONUT, a CIL method that relies on the combination of experience replay and contrastive learning. Through a modified version of the standard supervised contrastive loss applied only to the rehearsal samples, COCONUT preserves the learned representations by pulling closer samples from the same class and pushing away the others. Moreover, we leverage a multimodal contrastive loss that helps the model learn more discriminative representations of the new data by aligning audio and text features. We also investigate different contrastive designs to combine the strengths of the contrastive loss with teacher-student architectures used for distillation. Experiments on two established SLU datasets reveal the effectiveness of our proposed approach and significant improvements over the baselines. We also show that COCONUT can be combined with methods that operate on the decoder side of the model, resulting in further metrics improvements.

## 1 INTRODUCTION

With the rapid progress of intelligent voice-enabled personal assistants, the significance of Spoken Language Understanding (SLU) has gained substantial recognition in recent years (Arora et al., 2022; Qin et al., 2021). Conventional SLU models deploy a cascaded pipeline of an automatic speech recognition (ASR) system followed by a natural language understanding (NLU) module (Mesnil et al., 2014; Horlock & King, 2003). ASR maps the input speech into text representations, and NLU extracts the target intent labels from the intermediate text. Even though these approaches can leverage a vast abundance of ASR and NLU data, they suffer from ASR error propagation. Conversely, end-to-end (E2E) SLU (Agrawal et al., 2022; Lugosch et al., 2019; Saxon et al., 2021) has received more attention in recent research because it uses a single trainable model to map the speech audio directly to the intent labels, bypassing the need to explicitly generate a text transcript. This approach leads to reduced latency and error propagation.

The assumption that the data distribution the model will face after deployment aligns with what it encountered during the training phase is brittle and unrealistic. In fact, real-world scenarios entail evolving streams of data where novel categories (e.g., new vocabulary or intents) emerge sequentially, known as continual learning (CL). Unfortunately, while neural networks thrive in a stationary environment, the situation is reversed in CL, resulting in the "catastrophic forgetting" (CF) of the existing knowledge in favor of the fresh new information (McCloskey & Cohen, 1989). Although the majority of CL works have focused on computer vision tasks like image classification (Buzzega et al., 2020; Wang et al., 2022c) and semantic segmentation (Maracani et al., 2021; Yang et al., 2022a), a few works have recently turned their attention towards text (Wang et al., 2023a; Ke et al., 2023) and speech-related (Cappellazzo et al., 2023a; Diwan et al., 2023) problems, as well as vision-language (Ni et al., 2023; Zhu et al., 2023) and vision-audio (Mo et al., 2023; Pian et al., 2023).

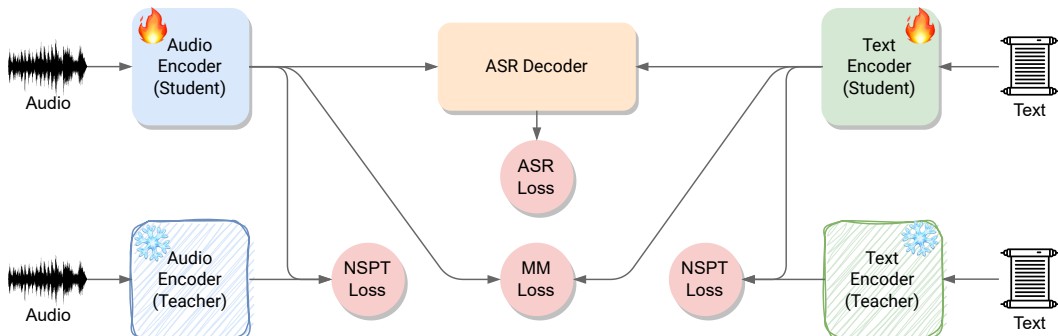

Figure 1: Overview of COCONUT: our proposed CL method. Aside from the standard ASR loss, COCONUT implements two contrastive learning-based losses. The NSPT (negative-student positive-teacher) loss is a supervised contrastive distillation loss that preserves the feature representations of the past classes for both audio and text samples. The positive and negative samples are computed with the teacher and student model, respectively. The MM (multi-modal) loss aims to align audio and text representations belonging to the same class. The combination of these two losses produces features that are more transferable and resilient to catastrophic forgetting.

While most SLU works have considered an offline setting, a thorough study of SLU under a class-incremental learning (CIL) setup still lacks. In CIL, one single model is adapted on a sequence of different tasks as incremental intent labels emerge sequentially. Recently, Cappellazzo et al. (2023b) studied the problem of CIL in ASR-SLU, where SLU is carried out in a sequence-to-sequence (seq2seq) fashion, thus computing the intent labels in an auto-regressive way together with the ASR transcriptions. By doing this, the model comprises three blocks: a text encoder, an audio encoder, and an ASR decoder. While Cappellazzo et al. (2023b) proposed to overcome CF by using knowledge distillation techniques applied to the ASR decoder, in this paper, we exploit the multi-modal audio-text setting and propose COCONUT: COntinual Contrastive spOken laNguage UndersTanding. COCONUT combines experience replay (ER) and contrastive learning principles. Whereas ER is a well-established approach in CL (Rolnick et al., 2019), only recently has contrastive learning been harnessed to learn representations continually. Both supervised (Cha et al., 2021; Yang et al., 2022a) and self-supervised (Fini et al., 2022; Wang et al., 2022c) contrastive learning have proven useful to lessen the CF issue. Specifically, COCONUT relies on two contrastive learning-based losses that operate on a shared embedding space where the audio and text features are projected.

The first contrastive loss, coined *Negative-Student Positive-Teacher* (NSPT), is a modified version of the supervised contrastive learning loss that aims to consolidate what the model has learned in the previous tasks. It also exploits the knowledge distillation principle (Hinton et al., 2015; Li & Hoiem, 2017) to guide the current model (student) to produce representations that follow the ones obtained with the model from the previous task (teacher). For this reason, this loss is computed only for the rehearsal data (i.e., the anchors). A key difference between our loss and the standard contrastive one is that the positive samples are computed using the teacher model (the positives only come from the rehearsal data), whereas the negatives are computed with the student. In this way, we avoid stale and scattered representations for the new data.

The second loss is inspired by the recent progress in multi-modal representation learning. Considering that for audio-text paired data, audio and text represent the same information but in different ways, it has been shown that aligning their representations results in better performance for various speech-related problems (Zhu et al., 2022; Ye et al., 2022; Manco et al., 2022). Therefore, we propose a multi-modal (MM) supervised contrastive loss that, exclusively applied to the current task's data, brings audio and text representations belonging to the same class into closer proximity in the shared feature space, resulting in features that are more transferable and resilient to CF. An overview of COCONUT is illustrated in Figure 1.

In summary, our contributions are the following:

- We introduce COCONUT, a CL method that makes use of two supervised contrastive learning objectives to mitigate catastrophic forgetting for seq2seq SLU models.

- We conduct extensive experiments on two popular SLU benchmarks and we show CO-CONUT achieves consistent improvements over the baselines. We also demonstrate that it can be combined with a KD technique applied to the ASR decoder, leading to further improvements.

- We finally ablate the contribution of each loss and its components, as well as the role of the temperature parameter in the contrastive continual learning process.

## 2 RELATED WORK

A vast array of CL strategies exist in the literature (Wang et al., 2023b; Zhou et al., 2023), which can be categorized into some macro groups: *regularization*-based, *experience replay*, and *architecture*-based. *Regularization* methods contrast forgetting either by introducing some ad-hoc regularization terms that penalize changes to model weights (Ebrahimi et al., 2019; Kirkpatrick et al., 2017) or to model predictions (Hou et al., 2018; Li & Hoiem, 2017; Fini et al., 2020). *Experience replay* approaches interleave the new data with cherry-picked samples from the prior tasks (Chaudhry et al., 2018; Bang et al., 2021; Buzzega et al., 2020), or they incorporate regularization terms with this additional data to steer the optimization process and prevent catastrophic forgetting (Chaudhry et al., 2018; Wang et al., 2021; Yang et al., 2022b). Finally, *architecture* methods involve creating task-specific/adaptive parameters, such as dedicated parameters to each task (Xue et al., 2022; Wang et al., 2022a) or task-adaptive sub-modules or subnetworks (Aljundi et al., 2017; Ostapenko et al., 2021).

Contrastive learning (Oord et al., 2018; Chen et al., 2020) is a popular approach in self-supervised learning, but it can also be used in supervised learning (Gui et al., 2023) and multimodal learning (Radford et al., 2021). Its objective is to learn discriminative feature representations by pushing apart different samples (negatives) and bringing closer similar ones (positives). In the case of supervised CIL, it has been shown that endowing the model with contrastive learning objectives results in more robust representations against CF. For incremental semantic segmentation, Yang et al. (2022a) and Zhao et al. (2023) propose to exploit contrastive learning in conjunction with knowledge distillation. For image classification, Wang et al. (2022b) advance a contrastive learning strategy based on the vision transformer architecture for online CL.

## 3 PROBLEM FORMULATION

### 3.1 ASR-SLU MULTI-TASK LEARNING

SLU is considered a more difficult task than ASR and NLU since it involves concurrent acoustic and semantic interpretation (Tur & De Mori, 2011). For this reason, it is common practice in the literature to include an additional ASR objective such that the intent and the transcript are generated in an auto-regressive fashion, resulting in a multi-task learning setting (Arora et al., 2022; Peng et al., 2023). By doing this, the text transcript input to the model includes a class intent token that is specific to the actual task.

Let $\theta$ be the parameters of a seq2seq ASR model, constituted by an audio encoder, a text encoder (i.e., embedding layer), and an ASR decoder. Let $\mathbf{x} = [x_0, \ldots, x_{U-1}]$ be an audio input sequence of length $U$, and $\mathbf{y} = [y_{cls}, y_{sep}, y_0, \ldots, y_{J-3}]$ be the corresponding "extended" input transcript of length $J$, where with the term "extended" we refer to the original transcript $[y_0, \ldots, y_{J-3}]$ augmented with the intent class token $y_{cls}$ and a special separation token $y_{sep}$. The goal of the ASR model is to find the most likely extended transcript given the input sequence $\mathbf{x}$:

$$\hat{\mathbf{y}} = \arg\max_{\mathbf{y} \in \mathcal{Y}^*} p(\mathbf{y}|\mathbf{x}; \theta), \tag{1}$$

where $\mathcal{Y}^*$ is the set of all token sequences. The predicted intent is obtained extracting $y_{cls}$ from $\hat{\mathbf{y}}$.

### 3.2 CLASS-INCREMENTAL LEARNING

For our experiments, we consider a CIL setting where we adapt a single model to learn sequentially $N$ tasks corresponding to non-overlapping subsets of classes (in our case *intents*). Put formally, the

training dataset is divided into $N$ distinct tasks, $\mathcal{D} = \{\mathcal{D}_0, \ldots, \mathcal{D}_{N-1}\}$, based on the intent token $y_{cls}$, so that one intent is included in one and only one task. The dataset $\mathcal{D}_n$ of task $n$ comprises audio signals $\mathcal{X}_n$ with associated transcriptions $\mathcal{Y}_n$, i.e. $\mathcal{D}_n = (\mathcal{X}_n, \mathcal{Y}_n)$. The CIL setting is challenging in that the model must be able to distinguish all classes until task $n$, thus at inference time the task labels are not available (unlike in task-incremental learning) (Hsu et al., 2018).

## 4 PROPOSED APPROACH

### 4.1 STANDARD REHEARSAL-BASED APPROACH

We assume the availability of a rehearsal buffer, $\mathcal{M}$, in which we can store a few samples for each class encountered in the previous tasks. During the training phase of task $n$, $\mathcal{D}_n$, we refer to $\mathcal{B}$ as a mini-batch of samples $(\mathbf{x}, \mathbf{y})$, some of which come from the current task and some from the rehearsal memory. To increase the variance of the audio data, we apply SpecAug (Park et al., 2019) to the audio waveform $\mathbf{x}$ as a data augmentation transformation. Regarding the transcript $\mathbf{y}$, we do not implement any augmentation technique. Then, we encode each modality separately through a dedicated feature encoder. An audio encoder maps each audio input into a feature vector $\mathbf{h}_A \in \mathbb{R}^{U \times d_A}$, where $d_A$ is the audio hidden size. Similarly, a text encoder converts each text input into a feature vector $\mathbf{h}_T \in \mathbb{R}^{J \times d_T}$, where $d_T$ is the text hidden size. At this point, if no specific CL losses are introduced, the ASR decoder generates the output sequence in an auto-regressive fashion, cross-attending on the audio encoder's feature representations $\mathbf{h}_A$. Therefore, at task $n$, we minimize the conventional cross-entropy loss over the current mini-batch $\mathcal{B}$:

$$\mathcal{L}_{\text{ASR}} = -\frac{1}{|\mathcal{B}|} \sum_{(\mathbf{x}, \mathbf{y}) \in \mathcal{B}} \log(p(\mathbf{y}|\mathbf{x}; \theta)). \tag{2}$$

### 4.2 COCONUT

**Preliminaries**. We introduce here some notations for the contrastive losses of COCONUT. Since we work on sequences of audio and text, we need to aggregate the features we obtain with the encoders before computing the contrastive loss. For the audio component $\mathbf{h}_A$ we apply a mean operation over its sequence length, whereas for text we only select the feature related to the intent token. Then, as is common practice in contrastive learning (Radford et al., 2021; Chen et al., 2020), the resulting embeddings go through two separate linear projection layers that map them into a shared embedding space. At inference time, the projection layers are discarded. Therefore, we get the projected embeddings $\mathbf{a}$ and $\mathbf{t}$ in the following way:

$$\mathbf{a} = g_A(avg(\mathbf{h}_A)), \quad \mathbf{t} = g_T(cls(\mathbf{h}_T)), \tag{3}$$

where $cls(\cdot)$ is a function that extracts the feature associated with the class token, $g_A(\cdot)$ and $g_T(\cdot)$ are the projection layers, $\mathbf{a} \in \mathbb{R}^{d_S}$ and $\mathbf{t} \in \mathbb{R}^{d_S}$, where $d_S$ is the dimension of the shared space.

Furthermore, we introduce some notations for the indices of samples coming from the current mini-batch $\mathcal{B}$. Let $\mathcal{I}_c$ and $\mathcal{I}_r$ represent the set of indices of the new task samples and the indices of the samples from the rehearsal memory (old task samples) in $\mathcal{B}$, respectively. Also, let $\mathcal{I} = \mathcal{I}_c \cup \mathcal{I}_r$, and we define $\mathcal{P}(k)$ as the set of indices of positive samples (i.e., samples with the same intent token).

The objective of a standard supervised contrastive loss (SCL) (Khosla et al., 2020) is to push the representations of samples with different classes (negative pairs) farther apart while clustering representation of samples with the same class (positive pairs) closely together. Suppose that we get from the projection layers a generic representation $\mathbf{z}_i^D$ for the $i$-th element in the batch, where $\mathbf{z} = \{\mathbf{a}, \mathbf{t}\}$ and the superscript $D$ denotes whether the representation is computed with the teacher or student model. A generic formulation of the SCL loss takes the following form:

$$\mathcal{L}_{\text{SCL}} = \sum_{k \in \mathcal{I}} \frac{-1}{|\mathcal{P}(k)|} \sum_{p \in \mathcal{P}(k)} \log \frac{\exp(\mathbf{z}_k^D \cdot \mathbf{z}_p^D / \tau)}{\sum_{i \in \mathcal{I}} \exp(\mathbf{z}_k^D \cdot \mathbf{z}_i^D / \tau)}, \tag{4}$$

where $\tau \in \mathbb{R}^+$ is a fixed temperature scaling parameter.

**Supervised Contrastive Distillation Loss (NSPT)**. This loss combines the benefits of knowledge distillation with those of contrastive learning. First of all, since the teacher model conveys information about the previous classes, we would like to use it as a guide for the student through a knowledge

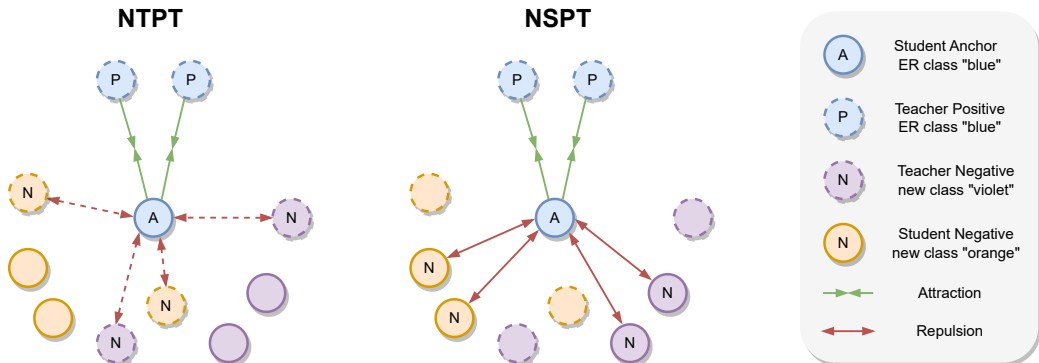

Figure 2: Illustration of the NTPT loss and our proposed NSPT loss. Given an anchor sample from the current mini-batch, the NTPT loss computes the negatives and positives using the teacher model (dashed circles). Instead, the NSPT loss, since the negative samples mainly come from the new classes and the teacher model has not been trained using those classes, computes the positives with the teacher while the negatives are computed with the student model (solid circles). If the features obtained with the teacher are scattered and static (the teacher is frozen), those obtained with the student are more clustered and can be learned during the current task. Best viewed in color.

distillation objective. In this way, the loss encourages the student model to produce audio and text embeddings consistent with those obtained by the teacher. Therefore, only the rehearsal samples are involved in this process as the teacher had no chance to see the current data. Additionally, we want to pull closer embeddings sharing the same intent class (i.e. the positives), while we push away the others (i.e. the negatives, whose class is different). This is obtained via a modified version of the standard supervised contrastive loss tailored for our setting. In fact, a standard one would use the teacher model to compute both the positives and the negatives (Khosla et al., 2020). However, since the teacher model is frozen and it is pointless to compute the representations of the samples from the current task using the teacher, we propose to use only the student model for computing the representations of the negatives. Therefore, our contrastive distillation loss computes the embeddings of the anchor and its corresponding negatives using the student model, while the positives come from the teacher (we call this loss *Negative-Student Positive-Teacher*, NSPT). On the contrary, for the standard contrastive loss both the positives and negatives are computed with the teacher (we call it *Negative-Teacher Positive-Teacher*, NTPT). Figure 2 illustrates visually how the NTPT and NSPT work in the shared embedding space. The NSPT loss is computed for both audio and text embeddings, leading to two components, one for each modality, as follows:

$$\mathcal{L}_{\text{NSPT}} = \sum_{k \in \mathcal{I}_r} \frac{-1}{|\mathcal{P}(k)|} \sum_{p \in \mathcal{P}(k)} \left[ \underbrace{\log \frac{\exp(\mathbf{a}_k^n \cdot \mathbf{a}_p^{n-1}/\tau)}{\sum_{i \in \mathcal{I}} \exp(\mathbf{a}_k^n \cdot \mathbf{a}_i^n/\tau)}}_{\mathcal{L}_{\text{A}}} + \underbrace{\log \frac{\exp(\mathbf{t}_k^n \cdot \mathbf{t}_p^{n-1}/\tau)}{\sum_{i \in \mathcal{I}} \exp(\mathbf{t}_k^n \cdot \mathbf{t}_i^n/\tau)}}_{\mathcal{L}_{\text{T}}} \right], \quad (5)$$

where $n$ and $n-1$ denote whether the representation is obtained with the student or teacher, and $\mathcal{L}_{\text{A}}$ and $\mathcal{L}_{\text{T}}$ represent the audio and text contribution, respectively. We empirically validate that the intuition of using negative samples from the student is beneficial in practice in section 5.3.

**Supervised Multi-Modal Contrastive Loss**. This loss is introduced for two reasons. First of all, since during the first task (no CL) the NSPT loss is not computed, this means that the projector layers of the model are not trained. This is a problem during the second task when the student distills the knowledge from a teacher with randomly initialized projectors. Second, we want to exploit the multi-modal nature of our SLU CIL setting. Consequently, we introduce a multi-modal (MM) loss that aims to align audio and text representations belonging to the same class, and thus training the projectors of the model. This alignment is achieved via a supervised multi-modal (i.e., audio-text) contrastive learning objective where feature representations of samples sharing the same intent token are attracted while the others are pushed away. Similar to (Kwon et al., 2022), we use the [CLS] text token ($y_{cls}$) for performing the multi-modal alignment. Furthermore, following (Cha et al., 2021), we always treat the rehearsal samples as negatives, preventing them from being anchors during the learning process. This design choice is buttressed by two motivations: 1) rehearsal data have been

learned by the previous model already and are preserved via the NSPT loss, and 2) we encourage the model to produce clusters for the new data that are separated from those of the rehearsal data. Formally, the MM loss takes the following form:

$$\mathcal{L}_{\text{MM}} = \sum_{k \in \mathcal{I}_c} \frac{-1}{|\mathcal{P}(k)|} \sum_{p \in \mathcal{P}(k)} \left[ \log \frac{\exp(\mathbf{a}_k \cdot \mathbf{t}_p / \tau)}{\sum_{i \in \mathcal{I}} \exp(\mathbf{a}_k \cdot \mathbf{t}_i / \tau)} + \log \frac{\exp(\mathbf{t}_k \cdot \mathbf{a}_p / \tau)}{\sum_{i \in \mathcal{I}} \exp(\mathbf{t}_k \cdot \mathbf{a}_i / \tau)} \right]. \tag{6}$$

The first term of the internal loss is the audio-to-text component, whereas the second is the text-to-audio component (Zhang et al., 2022). The presence of both directions ($A \rightarrow T$ and $T \rightarrow A$) makes the MM loss symmetric. All in all, COCONUT minimizes the following loss:

$$\mathcal{L} = \mathcal{L}_{\text{ASR}} + \lambda_{\text{MM}} \mathcal{L}_{\text{MM}} + \lambda_{\text{NSPT}} \mathcal{L}_{\text{NSPT}}, \tag{7}$$

where lambdas are loss-specific weights. An overview of COCONUT is illustrated in Figure 1.

## 5 EXPERIMENTS

### 5.1 EXPERIMENTAL SETUP AND IMPLEMENTATION DETAILS

**Datasets and CIL setting**. We evaluate COCONUT on two SLU datasets: the Fluent Speech Commands (FSC) (Lugosch et al., 2019) and the Spoken Language Understanding Resource Package (SLURP) (Bastianelli et al., 2020). FSC includes 30,043 English utterances, recorded at 16 kHz. It includes 31 intent classes in total. The SLURP dataset comprises around 56 hours of audio of people interacting with a home assistant (*slurp_real*), with the addition of 43.5 hours of synthetic data (*slurp_synth*). It is considered the most challenging SLU dataset due to its lexical complexity. Each utterance is annotated with 3 semantics: scenario, action, and entity. The pair (scenario, action) defines an intent. Overall, there are 18 scenarios and 69 intents. For our experiments, we only perform intent classification. Following (Cappellazzo et al., 2023b), we use the scenario labels as splitting criterion to define the CIL setting. We experiment on two configurations: 1) the datasets are partitioned into 3 tasks, each task comprising 6 scenarios for SLURP (denoted as SLURP-3), and 10 intents for FSC (FSC-3); 2) a more challenging configuration with 6 tasks, each task including 3 scenarios for SLURP (SLURP-6), and 5 intents for FSC (FSC-6).

**Implementation Details**. For both datasets, the text encoder is a standard text embedding layer with size 768. For the audio encoder, we use a base Wav2vec 2.0 model (Baevski et al., 2020) pre-trained and fine-tuned on 960 hours of Librispeech for SLURP ($\sim$ 94.3M parameters), while we use base DistilHuBERT (Chang et al., 2022) for FSC ($\sim$ 23.5M parameters). Since FSC is a less challenging dataset than SLURP, we found that a smaller pre-trained encoder is sufficient to achieve state-of-the-art results. Both encoders have hidden sizes of 768 and their feature extractor is kept frozen during training. As in (Radford et al., 2021), we employ linear projection layers to map from each encoder's representation to the audio-text embedding space, whose dimension is 512. The ASR decoder is transformer-based with 6 layers, hidden size equal to 768, 8 attention heads, and the dimension of the feedforward layers is 2048.

For the tokenization we apply Byte-Pair Encoding (BPE) (Sennrich et al., 2016) for SLURP, with a vocabulary size of 1000 and BPE dropout equal to 0.1, whereas for FSC, given the limited number of unique words, we use word tokenization, resulting in 139 tokens. BPE automatically assigns to each intent a dedicated token, whereas for FSC we manually add the intent tokens. We refer the reader to the appendix for an exhaustive description of the hyperparameters. Regarding the weight coefficients, we set $\lambda_{\text{MM}}$ to 0.1, and similar to (Douillard et al., 2022; Wu et al., 2019) we set $\lambda_{\text{NSPT}}$ to $\frac{L_p}{L_p + L_n}$, where $L_p$ and $L_n$ count the number of past and new classes.

**Baselines**. Apart from the standard **offline** (1 task, no continual) and **fine-tuning** (no CL strategies) baselines, we compare COCONUT against standard **experience replay** (ER) methods with *random* and *iCaRL* (Rebuffi et al., 2017) sampling strategies. We note that ER is already a strong baseline for FSC and SLURP. Additionally, we report two methods proposed in (Cappellazzo et al., 2023b): audio-KD (**A-KD**) that applies the KD on the audio features of the rehearsal samples, and seq-KD (**S-KD**) that at the end of the current task stores the text transcriptions computed with beam search for the rehearsal samples and use them as pseudo-transcriptions for the next task. This method operates on the ASR decoder. We also report text-KD (**T-KD**), the text counterpart of the A-KD.

Table 1: Results in terms of Average Accuracy (↑), Last Accuracy (↑), and Average WER (↓) for different strategies on FSC and SLURP datasets. The second column represents the experience replay (ER) buffer size and the selection strategy (Selec.) used to populate the buffer. **Bold** and underscore numbers denote the best and second best method for a specific setting and metric, respectively. We show in the last row that COCONUT and S-KD can be used together, leading to the best results. For simplicity, the values of the last row are not in bold even though attain the best results.

| Setting → | | FSC-3 | | | FSC-6 | | | SLURP-3 | | | SLURP-6 | | |
|---|---|---|---|---|---|---|---|---|---|---|---|---|---|
| Metric → 
 Method ↓ | ER size/ 
 Selec. | Avg 
 Acc | Last 
 Acc | Avg 
 WER | Avg 
 Acc | Last 
 Acc | Avg 
 WER | Avg 
 Acc | Last 
 Acc | Avg 
 WER | Avg 
 Acc | Last 
 Acc | Avg 
 WER |
| Offline | - | 99.28 | - | 0.48 | 99.28 | - | 0.48 | 84.41 | - | 17.65 | 84.41 | - | 17.65 |
| Fine-tuning | - | 49.13 | 17.61 | 36.37 | 29.92 | 7.59 | 54.66 | 46.65 | 18.42 | 28.32 | 31.90 | 10.57 | 34.79 |
| ER | 2% / rand | 88.02 | 84.20 | 9.19 | 81.19 | 79.71 | 13.75 | 75.62 | 68.68 | 19.55 | 72.75 | 68.49 | 22.98 |
| ER | 1% / rand | 79.17 | 69.81 | 15.87 | 68.61 | 63.71 | 24.04 | 71.44 | 61.88 | 21.25 | 66.57 | 58.22 | 24.50 |
| ER | 1% / iCaRL | 82.04 | 74.00 | 13.45 | 69.76 | 64.12 | 23.22 | 71.94 | 63.22 | 21.06 | 68.08 | 62.29 | 26.05 |
| T-KD | 1% / iCaRL | 82.11 | 75.43 | 12.95 | 69.08 | 64.73 | 23.82 | 72.44 | 62.43 | 21.19 | 66.95 | 60.47 | **24.26** |
| A-KD | 1% / iCaRL | 84.79 | 78.12 | 11.54 | 73.54 | 67.05 | 20.36 | 72.10 | 63.84 | **20.67** | 68.52 | 62.51 | 24.29 |
| S-KD | 1% / iCaRL | 84.29 | 75.31 | 12.39 | 73.65 | 67.71 | 21.27 | **74.28** | **65.95** | 21.26 | 69.91 | 63.22 | **24.26** |
| COCONUT | 1% / iCaRL | **86.39** | **80.21** | **11.08** | **76.45** | **73.80** | **19.05** | 72.75 | 64.62 | 21.25 | **70.17** | **63.66** | 24.29 |
| COCONUT +S-KD | 1% / iCaRL | 87.64 | 80.45 | 10.49 | 77.57 | 74.01 | 18.47 | 75.58 | 67.39 | 20.61 | 71.91 | 65.41 | 24.16 |

**Metrics**. Following (Douillard et al., 2022), we report the results in terms of the *Avg Acc*, which is the average of the intent accuracies after each training task, and the *Last Acc*, which is the intent accuracy after the last task. We also report the *Avg WER*, defined as the average of the Word Error Rate (WER) of the extended transcription after each task.

## 5.2 MAIN RESULTS

In the first two rows of Table 1, we include the upper and lower bounds represented by the offline learning (which is in line with the state-of-the-art) and fine-tuning approaches. For the fine-tuning approach, we can notice how CF deteriorates the knowledge of the prior classes (Last Acc). We then include ER baselines with buffer capacity equal to 1 or 2% of the dataset size. While all methods use 1%, we also include one with 2% to show how COCONUT and the other methods perform with respect to this one, but using half memory. From these results we can see that ER-based methods achieve good results for all metrics and configurations, confirming themselves as solid baselines. For FSC, COCONUT outperforms the other baselines by a significant margin, in terms of both accuracy and WER. Its combination with the S-KD leads to additional improvements (last row).

If we turn our focus on SLURP we see that, for the setting with 3 tasks, in terms of intent accuracy S-KD turns out to be the best approach, followed by COCONUT. For the WER, all the methods achieve similar performance and do not provide significant enhancements. We speculate that, as only some words are task-specific while the others are spread across multiple tasks, the text modality is less affected by CF. It is also compelling to note that the A-KD always achieves better performance than T-KD, a trend that will also be observed for the NSPT loss in the ablation study. For SLURP-6, COCONUT slightly surpasses S-KD in terms of accuracy, and performs on par with the others for the WER metric. This indicates that COCONUT scales properly with the number of tasks, where the setting becomes more challenging. Additionally, we point out that for SLURP COCONUT provides less noticeable improvements than FSC. This can be attributable to the higher complexity of the dataset due to its larger dictionary and to the larger number of intents with respect to FSC (69 vs. 31). Finally, similar to FSC, the combination of COCONUT with S-KD attains the best results, confirming that fighting CF both at the encoders and ASR decoder is an effective solution.

In Fig. 3 we illustrate the trend of the intent accuracy after each task for FSC-6 and SLURP-6, respectively. For FSC-6, COCONUT outperforms the other baselines by a large margin after each task. For SLURP-6, COCONUT has a similar trend as S-KD, and their combination leads to a noteworthy boost to such an extent that after task 3 it even beats the baseline that uses twice as much memory. On the left part of Fig. 4 we show the trend of the WER task by task. If it is evident that COCONUT and its combination with S-KD outstrip the other baselines, we can also observe that the gap between COCONUT and the baseline with 2% of rehearsal samples is more prominent for the WER than it was for the accuracy. On the right of Fig. 4, we study the trend of COCONUT

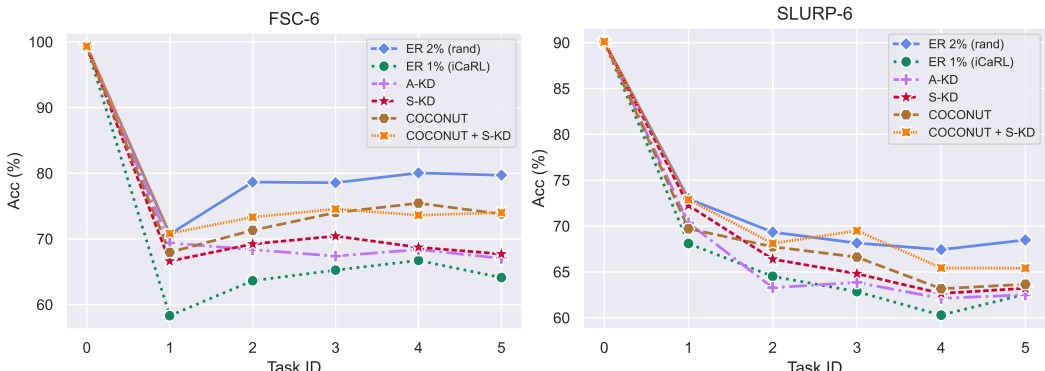

Figure 3: *Left*: the trend of the intent accuracy on the observed tasks for the FSC-6 setting. *Right*: the trend of the intent accuracy on the observed tasks for SLURP-6.

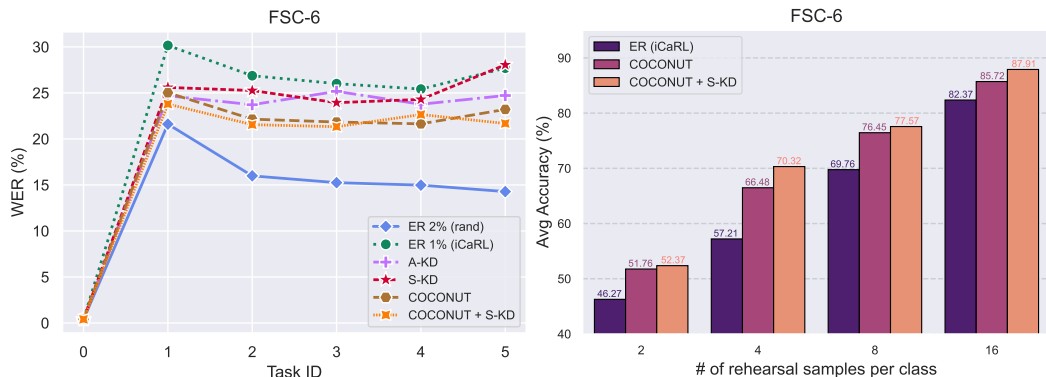

Figure 4: *Left*: the trend of the WER on the observed tasks for the FSC-6 setting. *Right*: the accuracy of COCONUT and other methods as a function of the memory size.

for different values of rehearsal samples per class. Note that 8 samples per class is tantamount to a buffer of capacity 1% with respect to the entire training dataset. The maximum gain is reached for 4 and 8 samples per class (9.27 and 6.69, respectively), while for the extreme cases of 2 and 16 samples, the gap is reduced. This is explained by the fact that when few samples are stored for each class, the effect of the NSPT loss is highly reduced given its reliance on the rehearsal data, whilst in the opposite case the abundance of rehearsal data makes the ER baseline already strong.

## 5.3 ABLATION STUDY

In this section, we ablate some design properties of COCONUT. In Tab. 2 we evaluate the difference in performance between the standard NTPT loss and our proposed NSPT. For FSC-6, the use of our proposed NSPT loss gives a considerable improvement over the NTPT loss, with the Last Acc metric being more enhanced. For SLURP-6, the trend is maintained, and now the NTPT even brings a small deterioration over the MM baseline in terms of Last Acc. Also, the MM loss alone contributes positively over the ER baseline for both settings. We recall that we do not study the individual contribution of the NSPT loss due to the issue of the random projectors of the teacher during the second task (see section 4.2). In Table 3 we study the design properties of the MM loss, and with its best configuration, we determine the individual contribution of the audio and text components to the NSPT loss. As was evident for the A-KD and T-KD, with the former giving more valuable results, here we also discover that the audio component is predominant. Plus, the concurrent use of both components brings a moderate increase in accuracy, and this is due to the alignment between audio and text obtained via the MM loss.

**On the impact of the temperature parameter**. In this section, we analyze the role of the temperature parameter in the CIL process for the MM loss (see Eq. 6) on the FSC-6 setting. We first try to

Table 2: Ablation on the use of NSPT vs. NTPT.

| Dataset → | FSC-6 | | | SLURP-6 | | |
|---|---|---|---|---|---|---|
| Metric → | Avg | Last | Avg | Avg | Last | Avg |
| Method ↓ | Acc | Acc | WER | Acc | Acc | WER |
| ER 1%/iCaRL | 69.76 | 64.12 | 23.22 | 68.08 | 62.29 | 26.05 |
| MM | 71.12 | 67.76 | 22.88 | 68.78 | 62.94 | 24.81 |
| MM + NTPT | 74.05 | 67.61 | 21.22 | 68.91 | 62.57 | 24.69 |
| MM + NSPT | **77.09** | **74.01** | **18.47** | **70.17** | **63.66** | **24.29** |

Table 3: Ablation study of the MM (upper part) and NSPT (bottom part) components. **CLS**: whether only the intent class token is used; **Anchor**: whether ER data are excluded from the anchors. $\mathcal{L}_A/\mathcal{L}_T$: whether the audio/text component of NSPT loss is used.

| CLS | Anchor | $\mathcal{L}_A$ | $\mathcal{L}_T$ | Acc |
|---|---|---|---|---|
| | | | | 70.10 |
| ✓ | | | | 70.49 |
| | ✓ | | | 71.09 |
| ✓ | ✓ | | | **71.12** |
| ✓ | ✓ | ✓ | | 76.84 |
| ✓ | ✓ | | ✓ | 73.11 |
| ✓ | ✓ | ✓ | ✓ | **77.09** |

Table 4: Ablation study of the temperature $\tau$ for the MM loss. We experiment on FSC-6 by setting $\tau$ beforehand and making it a learnable hyperparameter as is common practice in offline settings (Radford et al., 2021). The light-blue row corresponds to the value we used for our experiments.

| Metric → | Avg | Last | Avg |
|---|---|---|---|
| Temp. ($\tau$) ↓ | Acc | Acc | WER |
| 0.07 | 71.06 | 64.75 | **22.07** |
| 0.1 | **71.12** | **67.76** | 22.88 |
| 0.2 | 71.01 | 62.35 | 22.78 |
| Learnable | 69.05 | 66.33 | 24.57 |

set the value beforehand (0.07, 0.1, 0.2), and then we make the temperature a learnable hyperparameter (initial value is 0.07). Results are reported in Table 4. We can observe that $\tau = 0.1$ is the best configuration for the accuracy metric. Note that, however, the model does not seem very sensible to the temperature for the Avg Acc, whereas the Last Acc is more influenced. Since the Avg Acc does not change much across the three configurations, yet the Last Acc sways much more, this means that for $\tau = 0.1$ the model struggles more during the initial tasks, but it performs better towards the end of the learning process. On the other hand, learning $\tau$ task by task does not seem to be the right choice as the Avg Acc and WER metrics deteriorate with respect to the other three configurations where it is fixed. In fact, we observed that during the first tasks, the model is learning the optimal value for $\tau$ until it finds it (this value approximately lies in the range $0.134 - 0.142$). This initial transitional phase penalizes the accuracy of the first tasks, which in turn leads to a deterioration in the Avg Acc metric.

## 6 CONCLUSION

In this work, we study the problem of E2E SLU using a seq-2-seq model for class-incremental learning. In order to mitigate catastrophic forgetting we propose COCONUT, a CL approach that exploits experience replay and contrastive learning. On the one hand, it preserves the previously learned feature representations via an ad-hoc supervised contrastive distillation loss, on the other it contributes to aligning audio and text representations, thus resulting in more transferable and robust to catastrophic forgetting representations. We show that COCONUT outperforms the other baselines and that synergizes with other knowledge distillation techniques operating on the decoder side. We finally dissect the design choices of COCONUT through specific ablation studies, as well as the influence of the temperature parameter throughout the continual learning process.

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

# A   SUPPLEMENTARY MATERIAL

## A.1   HYPERPARAMETERS

We list the main hyperparameters used for our experiments in table 5. We also mention the number of epochs for each setting. For FSC-3, the number of epochs for each task is {40,30,30}, while

for SLURP-3 we have $\{40,25,25\}$. For FSC-6 and SLURP-6 we use $\{40,30,30,30,30,30\}$ and $\{40,25,20,20,20,20\}$ epochs, respectively. We finally note that we set lr $= 5 \cdot 10^{-4}$ for the text encoder, the ASR decoder and the classifier, while for the audio encoder we set a smaller learning rate, lr $= 5 \cdot 10^{-5}$, because it is pre-trained.

Table 5: Training hyperparameters for FSC and SLURP.

| Hyperparameter | FSC | SLURP |
|---|---|---|
| Batch Size | 32 | |
| Optimizer | AdamW | |
| $\beta_1$ | 0.9 | |
| $\beta_2$ | 0.98 | |
| $\epsilon$ | $10^{-6}$ | |
| lr | $5 \cdot 10^{-4}$ | |
| Weight Decay | 0.1 | |
| Tokenizer | Word Tok. | BPE Tok. |
| Beam Search width | 5 | 20 |
| Temperature $\tau$ | 0.1 | |

## A.2 SPECAUG DETAILS

In this section, we elaborate on the use of SpecAug for augmenting the audio input data. SpecAug (Park et al., 2019) is a popular augmentation technique that is applied directly on the log mel spectrogram of an audio signal, with the aim of making the model invariant to features deformation. In the original paper, they advance three different types of distortions: *time warping*, and *time* and *frequency masking*, where blocks of consecutive time steps and frequency channels are zero-masked, respectively. Since our audio encoders (i.e., DistilHuBERT and Wav2vec 2.0) work on the raw audio waveforms, SpecAug is not applicable by default. In order to circumvent this problem, we apply an approximated version of SpecAug directly to the raw waveform, as proposed in the SpeechBrain library (Ravanelli et al., 2021). We randomly drop chunks of the audio waveform (by zero-masking) and frequency bands (with band-drop filters). Unlike the SpeechBrain implementation, we do not apply speed perturbation. In more detail, with probability $0.5$ we randomly drop up to 2 frequencies, while with probability $0.5$ we randomly drop up to 3 chunks of audio whose length is sampled from a uniform distribution $\sim \mathcal{U}(0, 0.05 \cdot len(x))$, where $len(x)$ is the length of the considered audio waveform $x$.

## A.3 LIMITATIONS

Our work comes with some limitations. First of all, the number of suitable SLU datasets for defining a CIL setting is limited since few datasets provide enough intent classes. Then, we could not use batches larger than 32 owing to computational limitations, and it is known that contrastive learning benefits from larger batches.

## A.4 FUTURE WORK

COCONUT relies on two contrastive learning-based losses applied to the projections of audio and text encoders outputs. In principle, COCONUT could be exploited in other multi-modal settings such as audio-vision or vision-language. Therefore, it would be interesting to study whether CO-CONUT can be exploited in other different multi-modal scenarios. Also, since these settings usually involve a larger number of classes than ours, we would be able to test how COCONUT scales to the number of tasks.

