# OpenReview forum: "Continual Contrastive Spoken Language Understanding"
_ICLR.cc/2024/Conference — Submitted to ICLR 2024_

### Official Review · Reviewer_ZUcH · 2023-10-30

**Soundness:** 3 good
**Presentation:** 3 good
**Contribution:** 2 fair
**Rating:** 5
**Confidence:** 4

**Summary:**

This paper introduces a continual learning method, COCONUT, for spoken language understanding. This method uses two contrastive learning objectives in order to mitigate the catastrophic forgetting issue. COCONUT is evaluated on two popular SLU datasets, FSC and SLURP. Results show that COCONUT outperforms baselines when combined with some knowledge distillation techniques. A thorough ablation study is also presented to show the importance of the design of both contrastive losses.

**Strengths:**

**Originality**: The modified version of the proposed contrastive loss, Negative-Student Positive-Teacher loss, is novel.

**Quality**: The paper has presented detailed and thorough analysis of experimental results. However, the results of the proposed method is not strong enough.

**Clarity**: The writing of this paper is coherent.

**Significance**: The paper has some impact on the speech community, especially those working on spoken language understanding. The design of NSPT loss can be applied to other tasks, both within the speech domain and in other modalities.

**Weaknesses:**

(1) COCONUT does not outperform other baselines in a more challenging dataset (SLURP), limiting its contribution.

(2) The architectural design of COCONUT is rather complicated and onerous.

**Questions:**

What is the running time of COCONUT compared to other baselines? Does it cost significant more time to run COCONUT?

---

> ### Author Response · Authors · 2023-11-14
> **Rebuttal for reviewer ZUcH**
>
> We thank the reviewer for the valuable comments to our paper. We address each point below.
>
> **Weaknesses**
>
> > COCONUT does not outperform other baselines in a more challenging dataset (SLURP), limiting its contribution.
>
> For the SLURP dataset, while COCONUT performs worse than S-KD for the less challenging setting with 3 tasks, for the 6-task setup COCONUT slightly outperforms S-KD. As discussed in the paper, this indicates that COCONUT is more appropriate for settings involving more tasks, where the effect of catastrophic forgetting is exacerbated. Nevertheless, we think it is interesting that COCONUT can synergize and be combined with the S-KD method. As COCONUT and S-KD operate on two different parts of the model, that is the encoders and ASR decoder respectively, it seems natural to exploit both of them such that forgetting is mitigated at both levels. We speculate that for SLURP, since the batch size is limited and the number of classes is higher than FSC, having more tasks and thus fewer classes for each task can be more beneficial for the contrastive objectives of COCONUT.
>
> > The architectural design of COCONUT is rather complicated and onerous.
>
> We are not sure what the reviewer refers to with the term “complicated”. COCONUT combines supervised contrastive learning and knowledge distillation concepts tailored for the CL setting at hand. Indeed, it does not require more compute power than a standard contrastive loss, which is considered an efficient method to learn multimodal representations (i.e., CLIP [1]), but it requires an attentive selection of how to select the anchors, positives, and negatives. Moreover, we do not agree with the term “onerous” since the additional compute time added by integrating COCONUT is limited and reasonable, as explained below in more detail, and the contrastive learning approach is not considered an onerous objective by default. We finally emphasize that COCONUT does not introduce any modifications to the model itself (audio and text encoders, feature projectors, ASR decoder are fixed throughout the continual learning process), and neither does it introduce new parameters like other architectural CL strategies for each new emerging task.
>
> **Questions**
>
> > **Question 1**: What is the running time of COCONUT compared to other baselines? Does it cost significant more time to run COCONUT?
>
> The main difference between the ER baselines (ER, A-KD, S-KD) and COCONUT is that the former focus on the rehearsal data only, while COCONUT is applied to both the rehearsal data (NSPT loss) and the new data (MM loss), and so COCONUT requires an additional compute time due to the MM loss. Nevertheless, this additional time does not hinder its applicability as it is limited. Indeed, for the FSC-6 setting, the KD baselines require an additional 5/6 % of compute time with respect to the fine-tuning baseline, while COCONUT requires around 15%. For SLURP-3, the KD baselines require around 10/13% of additional compute time, whereas COCONUT requires around 35%. Undoubtedly COCONUT requires slightly more running time than the other KD baselines that are only applied to the rehearsal samples, but this overhead is minimal and consequently we believe this is not an issue for a practical scenario, considering also that COCONUT leads to much improved performance. Additionally, from a memory overhead point of view, COCONUT requires the storage of the rehearsal samples and a copy of the model from the previous task. These storage requirements are the same as the audio and text KDs. Instead, the S-KD approach, in addition to the aforementioned storage requirements, also necessitates the storage of the rehearsal text transcriptions generated with beam search from the previous task, thus increasing the requested memory overhead with respect to COCONUT.
>
>
> **References**
>
> [1] Alec Radford et al.,  Learning transferable visual models from natural language supervision, ICML, 2021.

---

> ### Author Response · Authors · 2023-11-21
> **Did our clarifications resolve the reviewer ZUcH's doubts/questions?**
>
> Dear reviewer ZUcH,
>
> We appreciated your time and effort in reviewing our paper. As the discussion phase nears its end, we would like to inquire if we have effectively addressed the issues raised in the initial review. We are receptive to further input and fully committed to making additional experiments or discussing other points if required. Thank you again!

---

> > ### Comment · Reviewer_ZUcH · 2023-11-21
> > **Update after author response**
> >
> > Thanks for the clarification and adding the content. I would keep the original score as I still believe the performance increment presented is marginal and the impact of this work is limited to a niche setting. The paper could be improved by showing the strength of COCONUT in broader multimodal scenarios or more tasks. \
> > That being said, I won't object accepting this paper if other reviewers advocate for it.

---

### Official Review · Reviewer_YNzV · 2023-10-30

**Soundness:** 4 excellent
**Presentation:** 2 fair
**Contribution:** 3 good
**Rating:** 6
**Confidence:** 3

**Summary:**

The paper proposes a contrastive learning framework to overcome catastrophic forgetting in the class-incremental learning setting of spoken language understanding, which contains two main losses. First, the paper modifies the standard supervised contrastive loss from Negative-Teacher Positive-Teacher to Negative-Student Positive-Teacher, while applying loss on rehearsal samples only. Second, a multimodal contrastive loss is also use to align the audio and text features.

**Strengths:**

1. The modification of loss function is interesting to mitigate catastrophic forgetting for seq2seq SLU models.
2. Experiments on two benchmarks and the ablation studies verify the effectiveness of proposed method over the previous baselines, as well as the two proposed losses.

**Weaknesses:**

The main weakness of this paper is the unclearness in text and the insufficient in experiments. Please see the Questions part for details.

**Questions:**

1. For the design of NSPD loss, 1) since the author highlights that the NSPD loss is conducted only on rehearsal samples, what if the loss is conduct on all samples?; 2) In equation 5, why i belongs to I instead of I_c? (according to the figure 2, it seems that the repulsion is applied on new class (i.e., I_c) samples).
2. For the design of multimodal loss, when the loss is applied, only in the first task (to initialize the projection) or in every task? I think the main motivation of this paper is to overcome catastrophic forgetting. However, the multimodal loss is more like a trick for better results instead of for catastrophic forgetting? By the way, is this paper the first work to apply multimodal loss on SLU tasks?
3. For the text encoder, in my understanding, it is just an embedding layer instead of a module? If this is the case, I think there is no need to call it text encoder. For example, this kind of embedding layer also exists in GPT-3.5. But it is common to call that GPT-3.5 has only a text decoder (decoder-only structure) instead of a text encoder (embedding layer) and an auto-regressive decoder. The text encoder in the paper is quite confusing. Please let me know if I understand mistakenly.
4. For the experiments, 1) More results are needed at different ER ratio (2-4%, 5-10%) to show that the proposed method can produce consistent improvement in different settings. 2) Just a question, why the results on FSC-6 in Table 1(76.46) and Table 2(77.09) are different?

---

> ### Author Response · Authors · 2023-11-14
> **Rebuttal for reviewer YNzV**
>
> We thank the reviewer for the valuable comments and questions. We are going to discuss each of them in detail below.
>
> **Questions**
>
> **Question 1**. **1)** First of all, we want to say that in CL, it is not univocal whether to use only the rehearsal samples or all the samples (rehearsal + new ones) in the distillation process. Some papers use only the rehearsal samples, others exploit both rehearsal and new ones. It seems that this choice depends on the task at hand. In our case, as we stated in the paper, we decided to use only the rehearsal samples because we believe it is not reasonable to use the teacher model for the new data since the teacher was not trained on these new data. To verify this choice, we ran two experiments on SLURP-6 and FSC-6 using all the data for the NSPT loss, and we share them below (R means rehearsal and it is our proposed version, R + N also includes the new (N) data). We can observe that the additional use of the new data leads to poorer performance, thus proving that using only the rehearsal samples is the best alternative. We plan to include this additional experiment in the supplementary material as we think it is compelling for the paper. **2)** I is correct in equation 5 as we treat as negatives both the samples coming from the new task (new classes) and the rehearsal samples in the current minibatch whose class differs from the current anchor. Thus, given a rehearsal sample from the current minibatch (the anchor), we push away the samples having a different class from these samples, either it comes from the rehearsal buffer or the new task. In Figure 2 we decided to avoid including rehearsal samples with different class labels to not complicate the figure. Following the reviewer’s advice, we will add a sentence in the figure’s caption explaining this.
>
> |Method|Avg Acc|Last Acc|
> |-|-|-|
> |**FSC-6**|||
> |NSPT (R)|**77.09**|**74.01**|
> |NSPT (R+N)|76.30|72.34|
> |**SLURP-6**||
> |NSPT (R)|**70.17**|**63.66**|
> |NSPT (R+N)|69.74|62.54|
>
> **Question 2**. We thank the reviewer for highlighting parts of the paper that might be unclear. In the following, we answer point by point. **1)** The MM loss is applied to all the tasks of the CL learning process. **2)** The MM loss is buttressed by two motivations: first, we need to train the projection layers from the very first task, otherwise the NSPT loss would use randomly initialized projections. Second, by aligning audio and text embeddings of the data belonging to the new task, we get more consistent embeddings in the shared audio-text space such that the NSPT loss can rely on better feature representations. **3)** We are not the first paper to apply multi-modal losses for SLU, but all the other papers using multi-modal approaches are studied under an offline i.i.d. setting. We cited some recent works that use MM contrastive objectives in an offline scenario, including a paper related to SLU (Yi Zhu et al., Cross-modal transfer learning via multi-grained alignment for end-to-end spoken language understanding, Interspeech 2022).
>
> **Question 3**. We call it “text” encoder as it produces a feature representation for the text data, in a similar way as the audio encoder does for the audio counterpart. In ASR it is quite common to exploit a simple embedding layer as a text encoder. In our preliminary results, we tried to use more complex text encoders such as BERT, but we did not observe valuable improvements in response to the large overhead required by employing a big pre-trained encoder like that. So the term “encoder” has to be interpreted as the module that produces the representations that will be fed to the ASR decoder and to the projection layers. We will clarify this point in the paper as we think it is important.
>
> **Question 4**. **1)** Regarding the additional experiments with higher ER ratios such as 4%-10%, we note that when we increase the ER ratio we are implicitly moving towards the offline setting (the upper bound), and this is something we do not desire because catastrophic forgetting is highly reduced. As we can see from the bar plot in Figure 4, when we use an ER ratio of 2% (16 samples) the results we get start approaching the offline upper bound. With this in mind, we believe that ER is a viable CL method only if a limited ratio is used, preserving the conditions a CL setting abides by. Nevertheless, we ran some additional experiments (see table below) where we used 4% of rehearsal data (30 samples) for FSC-6 setting. As we can observe, COCONUT and COCONUT + S-KD still outperform the ER baseline when 4% of memory is used, confirming the trend for smaller ER ratios. At the same time, the Avg and Last accuracies are saturating towards the upper bound. This is why we chose to use 1% as the ER ratio since we think it is a reasonable value to test our proposed approach. **2)** We thank the reviewer for pointing out this typo. The correct value is 77.09. We forgot to update the Avg Acc value for COCONUT for FSC-6 in Table 1.

---

> > ### Author Response · Authors · 2023-11-14
> > **Rebuttal for reviewer YNzV (part 2)**
> >
> > We include here the table for the question 4.1 since we ran out of characters in the previous comment.
> >
> > | **Samples per class** | **2** | **4** | **8** | **16** | **30** |
> > | ------ | ------ | ------ | ------ | ------ | ------ |
> > |ER | 46.27 | 57.21 | 69.76 | 82.37 | 89.60 |
> > | COCONUT |  51.76 | 66.48 | 76.45 | 85.72 | 93.41 |
> > | COCONUT + S-KD | 52.37 | 70.32 | 77.57  | 87.91 | 93.93 |

---

> ### Author Response · Authors · 2023-11-21
> **Did our clarifications resolve the reviewer YNzV's doubts/questions?**
>
> Dear reviewer YNzV,
>
> We appreciated your time and effort in reviewing our paper. As the discussion phase nears its end, we would like to inquire if we have effectively addressed the issues raised in the initial review. We are receptive to further input and fully committed to making additional experiments or discussing other points if required. Thank you again!

---

> > ### Comment · Reviewer_YNzV · 2023-11-22
> > **Thanks for your response**
> >
> > The author response address most of my unclearness. So I have increased my score to 6.
> >
> > However, I agree with the opinion of Reviewer eVfc and ZUcH that the paper is somewhat limited on tasks and settings.

---

### Official Review · Reviewer_eVfc · 2023-10-31

**Soundness:** 3 good
**Presentation:** 3 good
**Contribution:** 3 good
**Rating:** 6
**Confidence:** 3

**Summary:**

This paper addresses spoken language understanding (SLU) in a continual learning setting. End-to-end joint ASR-SLU approach is used (no cascade).
A new approach called COCONUT is presented that uses both experience replay and contrastive learning losses (NSPT: a contrastive KD loss and MM a multimodal loss that aligns audio-text representations).
Experiments are made on two SLU benchmarks: SLURP and FSC; the exact SLU task is intent classification. The continual learning setting used is the one from (Capellazzo & al 2023). Experiments show that COCONUT can compete with experience replay  (ER) of buffer capacity of 1% (but is worse than ER with buffer capacity 2%)

**Strengths:**

-a new approach for SLU in a CL setting that is better than a strong experience replay (ER) benchmark

-experiments on 2 popular SLU benchmarks that demonstrate the effectiveness of the proposed appproach

**Weaknesses:**

-more details on continual learning setting used would have been welcome (ref to (Capellazzo & al 2023) is not very self-explanatory)

-experience replay (ER) baseline with buffer capacity of 2% is still better than COCONUT and it is unclear how using twice memory (2% instead of 1%) is a real bottleneck in real applications (authors could have commented this more)

**Questions:**

-how COCONUT could be adapted to more speech tasks ?

-experience replay (ER) baseline with buffer capacity of 2% is still better than COCONUT and it is unclear how using twice memory (2% instead of 1%) is a real bottleneck in real applications (authors could have commented this more)

---

> ### Author Response · Authors · 2023-11-14
> **Rebuttal for reviewer eVfc**
>
> We thank the reviewer for the valuable comments. We are going to discuss each of them in detail below.
>
> **Weaknesses**
>
> > "more details on continual learning setting used would have been welcome (ref to (Capellazzo & al 2023) is not very self-explanatory)"
>
> Regarding some missing details for the CL setting, we included the fundamental information for its definition in the first paragraph of section 5.1. We tried to include the most important details, but some details about the SLURP setting have been omitted for lack of space and because its definition is quite cumbersome. However, we agree with the reviewer that it is important and useful to include these additional details about the definition of the CL setting for the SLURP dataset in the supplementary material so that the reader can dive into more details. In the following, we elaborate a bit more on the CL setting defined for SLURP. As the SLURP dataset provides multiple levels of annotations (scenario, action, entity[es]), in principle one could decide to divide the dataset into tasks following one of these criteria. However, in the original paper where a CIL setting for SLURP is defined [1], the authors propose to use the scenarios as splitting criterion because they represent more general concepts than the actions and entities, and then the accuracy is computed on the intent, defined as the pair (scenario,action). As you may understand from this brief explanation, the definition of such a setting is pretty involved, and we thought it was not necessary to weigh down the flow of the experimental section. In addition to this, we mention that, following [1], we define the order of the classes in the various tasks depending on their cardinality, meaning that the classes with more samples are seen first by the model. This is done because the cardinality of SLURP scenarios varies consistently from class to class. We finally underline that we tried to be as consistent with the original implementation of [1] as possible.
>
> > experience replay (ER) baseline with buffer capacity of 2% is still better than COCONUT and it is unclear how using twice memory (2% instead of 1%) is a real bottleneck in real applications (authors could have commented this more)
>
> ER with 2% performs better than COCONUT with 1%, but in principle, it is not fair to compare two approaches that use a different amount of rehearsal data. Indeed, from the right figure of Table 2 we can see that if COCONUT also uses 2% of rehearsal data, it outperforms the ER baseline by a large margin. By including the ER baseline with 2% in Table 1 we wanted to stress the fact that COCONUT + S-KD for some settings (i.e., FSC-3, SLURP-3) manages to approach the ER baseline that uses twice as many rehearsal samples.
>
> **Questions**
>
> > **Question 1**: how COCONUT could be adapted to more speech tasks ?
>
> We are not sure whether the reviewer with the term “speech tasks” refers to other speech “problems” or more CL tasks. Unfortunately, in CL the term task is a bit ambiguous in that it can refer to both the two mentioned concepts. Regarding the first possibility, COCONUT has been designed to work in a multimodal setting, in our case audio-text. Nevertheless, the NSPT loss can be used in an unimodal setting by including only the audio or text component. Since the NSPT loss, and also the MM loss, only requires as input the output of an audio/text encoder, it can be used in any audio or speech classification task, such as keyword spotting or speaker identification. In our paper, we found that the audio component has a larger impact on the CL process than the text counterpart, while for other tasks this might not be the case. Additionally, COCONUT requires only two parameters to be tuned: the hidden dimension of the projected layer and the temperature parameter. However, these parameters are standard tunable parameters for any contrastive learning loss and we found that standard values worked well also in our CL setting. We also emphasize that we believe COCONUT is a promising approach for other multimodal settings such as audio-vision and language-vision tasks, as well as other audio-language tasks like audio-text retrieval. Finally, if the mentioned tasks are classification ones, our joint ASR-SLU problem is more challenging as the ASR decoder generates the final transcription in a seq2seq way, thus making our setting more specific. For the second interpretation, we have seen from the SLURP results that COCONUT performs better when the CL setting includes more tasks, so we believe COCONUT can scale to more tasks efficiently.
>
> **References**
>
> [1] U. Cappellazzo, M. Yang, D. Falavigna, and A. Brutti. Sequence-level knowledge distillation for class-incremental end-to-end spoken language understanding, Interspeech, 2023.

---

> > ### Comment · Reviewer_eVfc · 2023-11-19
> > **answer to authors**
> >
> > by 'more speech tasks' i meant beyond SLU (speech translation for instance) ; other than that tks for your answers on the ER buffer capacity...

---

### Official Review · Reviewer_6YBX · 2023-11-01

**Soundness:** 3 good
**Presentation:** 3 good
**Contribution:** 3 good
**Rating:** 6
**Confidence:** 3

**Summary:**

- The paper targets end-to-end SLU using a seq2seq style of model.
- This work aims to address problems of balancing efficient and performant continual learning of new tasks (intents) without the effects of catastrophic forgetting.
- To these ends, they introduce a model architecture and set of losses to take advantage of student-teacher network training, contrastive learning, and multi-modal (speech/text) alignment.
-  The approach (COCONUT) is applied to two datasets, FSC and SLURP, where the approach is highlighted.
- COCONUT is shown to outperform other methods in almost all cases, except when combined with the next best approach (S-KD) or when using a larger memory for ER.
- Ablation shows impact of memory size on effect of COCONUT vs ER.

**Strengths:**

Clear presentation of motivations behind the combination of losses, the decisions behind whether to use student vs teacher examples, etc.  Nice figures and appropriate complexity to educate without losing the goal of the paper in the weeds.

**Weaknesses:**

For readers that may not be as familiar with results of other SLU work (both E2E and non-E2E), inclusion of results from other work could be useful. Or if such comparisons are not fair, perhaps a note in the table to that effect. The text mentions the other work which describes those rows (like S-KD), but it could be nice to see numbers from other work itself as well (?) for clearer context as well as results of conventional SLU approaches that are not E2E.

**Questions:**

It may be possible to better include results from prior work and non-E2E approaches to give more context to the results within the tables.

Also, one grammar note, I believe "sensible" should be "sensitive" here in the last paragraph of section 5.3:
          Note that, however, the model does not seem very sensitive to the temperature for the Avg Acc, whereas the Last Acc is more influenced.

---

> ### Author Response · Authors · 2023-11-14
> **Rebuttal for reviewer 6YBX**
>
> We thank the reviewer for the valuable comments to our paper. We address each point below.
>
>  **Weaknesses**
> > "...inclusion of results from other work could be useful... it could be nice to see numbers from other work itself as well (?) for clearer context..."
>
> We do understand that it would be nicer to have more works to compare our approach with, but unfortunately, only the approaches in [1] studied our same setting, and indeed we have reported them in our study (A-KD, T-KD, S-KD). The A-KD and T-KD rely on the concept of knowledge distillation to guide the current model to produce consistent AUDIO and TEXT feature representations with respect to the teacher model, respectively. Similar to our work, they focus on the audio and text encoders to mitigate catastrophic forgetting. The S-KD approach, instead, concentrates on the ASR decoder by promoting continuity between the teacher and student models in terms of the transcriptions they produce. Note that these methods combine knowledge distillation and experience replay strategies, which are two of the main families of continual learning approaches. For the SLURP dataset, almost all works treat the SLU problem as a multi-task ASR-SLU model where the labels are predicted along with the text transcriptions. This is because the SLURP dataset is a recent dataset and in the last couple of years E2E models have taken over the SLU domain (SpeechBrain, ESPNet, and NeMO recipes for SLURP all use E2E joint ASR-SLU approaches). For non-E2E approaches, we are not sure how COCONUT could be integrated as the NLU block works only on the text data produced by the ASR block, while in our case the audio and text data are exploited and combined using an E2E approach.
>
> **Questions**
>
> > **Question 1**:  "It may be possible to better include results from prior work and non-E2E approaches to give more context to the results within the tables."
>
> Unfortunately, the only prior work that tackled the problem of joint ASR-SLU under a CIL setting is [1], so we are not able to include other works/approaches apart from this one.
>
> > **Question 2**: "Also, one grammar note, I believe "sensible" should be "sensitive"..."
>
> We thank the reviewer for pointing out this typo, the correct term is “sensitive” and not “sensible”. We have corrected it.
>
> **References**
>
> [1] U. Cappellazzo, M. Yang, D. Falavigna, and A. Brutti. Sequence-level knowledge distillation for class-incremental end-to-end spoken language understanding, Interspeech, 2023.

---

### Author Response · Authors · 2023-11-22
**Final remarks**

Dear reviewers and ACs,

Let us first thank the reviewers for their valuable time and engagement in the discussion. We extremely appreciated the comments and the various suggestions/questions, we believe they have greatly helped to improve the article. In the end, we would like to elaborate a bit on two aspects/weaknesses raised by multiple reviewers: 1) **allegedly limited and niche setting**, and 2) **extension of COCONUT to more tasks**.

**1)** In this paper, we studied the problem of class-incremental learning applied to a joint Spoken Language Understanding-Automatic Speech Recognition (SLU-ASR) setting. The problems of SLU and ASR are crucial in all speech-related tasks where a device needs to process and elaborate the speech signal and perform some sort of decisions, such as voice-assistants and smart home devices (e.g., Siri, Alexa, etc.). These systems must be able to integrate novel concepts like vocabularies over time without retraining from scratch. Thus, while we strongly believe that it is compelling to endow these systems with lifelong learning capabilities, we also think that this setting is more challenging than a standard mainstream class-incremental learning one because *1)* it includes the mutual interaction of two modalities, text and audio, and *2)* aside from the standard feature encoder + classifier pipeline, our setting also includes the ASR decoder, which is affected by catastrophic forgetting as well. The speech tasks that abide by these constraints are limited because we need at the same time both audio and text data, and a label associating each data pair to the corresponding class to define the class-incremental learning setting. While some tasks such as keyword spotting and emotion recognition do not meet these criteria, SLU + ASR, instead, is a suitable task. As pointed out by reviewer *eVfc*, speech translation could be an interesting additional task to consider, even though each task would entail a different language, so it would be quite different from our class-incremental learning setup, and its definition not straightforward. For this reason, we decided to focus entirely on joint SLU-ASR, which has a big impact on many everyday applications.

**2)** We also understand that it would be interesting and useful to test *COCONUT* on other multi-modal tasks like vision-language and audio-visual tasks, but at the same time we think that the definition of such settings is as complex as that of our audio-language setting, so we speculate it could complicate the flow of the paper and lose the motivations for which we studied it (these multimodal tasks use feature encoders + classifier, the ASR decoder is not involved). Therefore, we decided to keep it for future work.

All in all, we hope the reviewers appreciated our effort to study a more challenging and under-explored class-incremental learning setting, offering a new perspective on it and laying the ground for future extensions and for people willing to delve into this task.

---

### Meta-Review · Area_Chair_G5bC · 2023-11-30

**Metareview:**

This paper proposes COCONUT, an end-to-end joint ASR-SLU (automatic speech recognition-spoken language understanding) approach, to address continual learning where intent classes are added / changed over time. The work introduces two contrastive learning objectives to alleviate catastrophic forgetting. Authors show the effectiveness of COCONUT empirically by evaluating on FSC and SLURP, two SLU benchamarks, where COCONUT outperforms baseline, experience replay, in most settings.

Strength:
 - Clear presentation and illustration for proposed approach and results
 - A new and interesting approach for SLU in continuous learning. Approach seems effective based on the experiments done in the paper. This can have reasonable impact on the SLU community.

Weakness: as multiple reviewers pointed out, the paper works on a niche area (continuous learning for SLU). This can also be told that there is very limited baseline reported in this work, and the baseline comes from similar set of authors. Sometimes it's fine to work on a niche area if the result/algorithm has potential to be generalized, or the problem formulation etc. is innovative and ground breaking. However, I think this paper is neither of the case. The algorithm seems tailored towards SLU problems. The author mentioned in the final remarks even porting the algorithm to speech translation, which seems relevant enough to ASR, can take non-trivial efforts, not even mentioned other modalities. The problem formulation is not very new (applying continuous learning to a new task seems marginally novel). Considering all these factors, I think this work is of limited interests to ICLR community. This work is more suitable for a speech focused venue.

**Justification For Why Not Higher Score:**

as mentioned in the weakness, this work focuses on a very niche area and generalizing the method/results seems not straightforward. The work was only evaluated on two datasets in SLU. All of these suggest that this work has limited impact and is not suitable for ICLR.

**Justification For Why Not Lower Score:**

N/A

---

### Decision · Program_Chairs · 2024-01-16

Reject